# Increased Trypanocidal Activity of the Salinomycin Derivative Ironomycin Is Due to ROS Production and Iron Uptake Impairment

**DOI:** 10.3390/molecules29235597

**Published:** 2024-11-27

**Authors:** Dietmar Steverding, Stuart A. Rushworth, Georgina R. Hurle, Michał Antoszczak, Michał Sulik, Adam Huczyński, Kevin M. Tyler

**Affiliations:** 1Bob Champion Research and Education Building, Norwich Medical School, University of East Anglia, Norwich NR4 7UQ, UK; s.rushworth@uea.ac.uk; 2BioMedical Research Centre, Norwich Medical School, University of East Anglia, Norwich NR4 7TJ, UK; g.hurle@uea.ac.uk (G.R.H.); k.tyler@uea.ac.uk (K.M.T.); 3Department of Medical Chemistry, Faculty of Chemistry, Adam Mickiewicz University, Uniwersytetu Poznańskiego 8, 61-614 Poznań, Poland; michal.antoszczak@amu.edu.pl (M.A.); michal.sulik@amu.edu.pl (M.S.);

**Keywords:** *Trypanosoma brucei*, ironomycin, salinomycin, trypanocidal activity, reactive oxygen species

## Abstract

Salinomycin and its derivatives display promising anti-proliferating activity against bloodstream forms of *Trypanosoma brucei*. The mechanism of trypanocidal action of these compounds is due to their ionophoretic activity inducing an influx of sodium cations followed by osmotic water uptake, leading to massive swelling of bloodstream-form trypanosomes. Generally, higher trypanocidal activities of salinomycin derivatives are associated with higher cell swelling activities. Although ironomycin (C20-propargylamine derivative of salinomycin) and salinomycin showed identical cell swelling activities, ironomycin was 6 times more trypanocidal than salinomycin, and the 50% growth inhibition (GI_50_) values were 0.034 μM and 0.20 μM, respectively. However, when bloodstream-form trypanosomes were incubated with ironomycin in the presence of vitamin E and ammonium ferric citrate, the trypanocidal activity of the compound was reduced to that of salinomycin (GI_50_ = 0.21 μM vs. GI_50_ = 0.20 μM). In addition, vitamin E was found to decrease the trypanocidal activity of ironomycin much more than ammonium ferric citrate (GI_50_ = 0.18 μM vs. GI_50_ = 0.042 μM). Moreover, ironomycin caused a reduction in the uptake of the iron-carrier protein transferrin mediated by a downregulation of the transferrin receptor and led to the accumulation and sequestering of iron(II) in the parasite’s lysosome, triggering an increase production of reactive oxygen species (ROS). These results suggest that the increased trypanocidal activity of ironomycin can be mainly attributed to an increased ROS production and, to a lesser extent, an impairment in iron uptake.

## 1. Introduction

African trypanosomiasis is a neglected tropical disease affecting both humans and livestock animals. The etiological agents of the disease are protozoan parasites of the genus *Trypanosoma*. Human African trypanosomiasis (sleeping sickness) is caused by *T. brucei* ssp. whereas animal African trypanosomiasis (nagana disease) by *T. brucei* ssp., *T. congolense*, and *T. vivax*. While the reported number of cases of sleeping sickness has dropped below 1000 since 2018 [1], nagana disease remains a major problem in sub-Saharan Africa with around 3–18% of the over 500 million heads of cattle, sheep, and goats infected [2]. Although many sub-Saharan countries seem to have been freed from sleeping sickness [2], the presence of animal African trypanosomiasis represents a reservoir for re-introducing the disease in humans. Therefore, it is still necessary to continue the effort to develop new trypanocidal drugs, especially to control nagana disease, as a One Health approach to achieving the elimination of African trypanosomiasis [3].

Polyether ionophores, in particular salinomycin and its derivatives, have been shown to inhibit the growth of culture-adapted bloodstream forms of *Trypanosoma brucei* [4,5]. The trypanocidal mode of action of salinomycin was determined to involve the transport of sodium cations across the plasma membrane into bloodstream-form trypanosomes [6]. The increased intracellular sodium concentration subsequently causes an influx of water leading to massive cell swelling of the trypanosomes [6]. Although salinomycin derivatives have been shown to induce cell swelling in trypanosomes, the intensity of the cell swelling depends on the modification [4,5]. Salinomycin derivatives with increased trypanocidal activity generally show increased cell swelling activity [4,5].

Ironomycin, a C20-propargylamine derivative of salinomycin (Figure 1), has been previously demonstrated to be 10 times more potent in inhibiting the cell growth of cancer stem cells in vitro and in vivo than the parent compound salinomycin [7]. The reason for this is that ironomycin exhibits more potent and selective activity against cancer stem cells than salinomycin. Although both ionophores accumulate in the lysosome, salinomycin also targets the endoplasmic reticulum, Golgi apparatus, and mitochondria [7,8]. Both compounds were found to block the translocation of iron(II) from the lysosome to the cytosol and to interact with iron(II), but ironomycin associates with the metal more potently [7,8]. These findings indicate that ironomycin is a stronger lysosomotropic agent and sequesters iron more efficiently in lysosomes than salinomycin. The pKa values of the ionophores support this suggestion. With a pKa value of 7.2, ironomycin accumulates more readily in the acidic lysosome than salinomycin with a pKa value of 6.5. The accumulation of iron(II) in the lysosome leads to the generation of reactive oxygen species (ROS) via Fenton chemistry. As a result of ROS production, the lysosomal membrane is peroxidized and subsequently becomes permeable causing cell death via ferroptosis [7].

Similarly to cancer stem cells, we found that ironomycin was 6 times more trypanocidal than salinomycin [5]. However, both compounds were found to exhibit identical ionophoretic activities as they both induce the same cell swelling rates in bloodstream-form trypanosomes [5]. Therefore, we were interested in investigating the cause of the increased trypanocidal activity of ironomycin and whether this was due to an impairment of iron homeostasis and ROS production.

## 2. Results and Discussion

### 2.1. Confirmation of Trypanocidal and Cell Swelling Activity of Ironomycin and Salinomycin

First, we conducted experiments to confirm our previous findings that salinomycin and ironomycin display different anti-trypanosomal activities but identical cell swelling activity. Changes in cell volume of bloodstream-form trypanosomes were determined using the light scattering method described before [6]. As shown in Figure 2, the cell swelling activity of ironomycin was indistinguishable from that of salinomycin. Over the whole measurement period of 1 h, no significant difference was observed in the cell swelling rates between salinomycin and ironomycin.

The trypanocidal activity of salinomycin and ironomycin was determined in vitro using the resazurin cell viability assay [9]. Both compounds showed a dose-dependent inhibitory effect on the growth of bloodstream forms of *T. brucei*, with 50% growth inhibition (GI_50_) values of 0.20 ± 0.02 μM and 0.034 ± 0.002 μM, respectively (Table 1 and Figure 3a). These values are very similar to the previously determined GI_50_ value for salinomycin (0.21 ± 0.06 μM) and ironomycin (0.035 ± 0.005 μM) [5], confirming that the C20-propargylamine derivative is about 6 times more trypanocidal than its parent compound.

### 2.2. Effect of Ironomycin on Iron Uptake

Next, we investigated whether the increased trypanocidal activity of ironomycin is due to the compound’s ability to interfere with cellular iron metabolism. Although ironomycin and salinomycin have been shown to sequester iron in lysosomes, leading to cytoplasmic iron depletion in cancer stem cells, ironomycin seems more potent in this action than salinomycin [7]. The reason for this may be attributed to the fact that ironomycin is stable in the presence of iron while salinomycin is slowly degraded by the metal [7]. To see whether iron supplementation can reduce the trypanocidal activity of ironomycin, the effect of 50 μM of ammonium ferric citrate on the anti-trypanosomal activity of both ironomycin and salinomycin was determined. (It should be noted that the amount of free iron in the medium is extremely low (<10^−18^ M). This is because transferrin supplemented with the bovine serum restricts the free iron concentration in the medium to 10^−18^ M [10].) Although ammonium ferric citrate reduced the trypanocidal activity of ironomycin slightly as evidenced by an increase in the GI_50_ value from 0.034 ± 0.002 μM to 0.042 ± 0.002 μM (*p* = 0.0046), the iron salt increased the trypanocidal activity of salinomycin, as demonstrated by a decrease in the GI_50_ value from 0.20 ± 0.02 μM to 0.15 ± 0.02 μM (*p* = 0.0105) (Table 1 and Figure 3a). This result suggests that part of the trypanocidal activity of ironomycin is due to iron depletion, which can be overcome by iron supplementation. Considering that the addition of iron may itself cause toxicity as indicated by the increased trypanocidal activity of salinomycin in the presence of ammonium ferric citrate, the effect of iron supplementation on the reduction of the trypanocidal activity of ironomycin may be even greater. Indeed, at a concentration of 50 μM, ammonium ferric citrate impaired the growth of bloodstream-form trypanosomes by about 10% (Appendix B, Figure A2). In addition, it is most likely that the added iron was not directly taken up by bloodstream-form trypanosomes but via transferrin that bound the iron from the ammonium ferric citrate included in the culture medium. It should be noted that the iron saturation of transferrin is usually only 30% [11]. It is reasonable to assume that if more iron-saturated transferrin (holo-transferrin) is available, bloodstream-form trypanosomes can get more iron to compensate for the iron depletion caused by ironomycin. To prove this assumption, the trypanocidal activity of ironomycin was tested in the presence of 1.2 mg/mL holo-transferrin, a concentration that is 10 times higher than the level of the iron-carrier protein in the culture medium (120 μg/mL [12]). Holo-transferrin reduced the trypanocidal activity of ironomycin more than ammonium ferric citrate (Table 1 and Figure 3), as evidenced by an increase in the GI_50_ value to 0.051 ± 0.005 μM (Table 1 and Figure 3b). Importantly, the addition of 1.2 mg/mL iron-free transferrin (apo-transferrin) did not affect the trypanocidal activity of ironomycin, as demonstrated by an unchanged GI_50_ value of 0.034 ± 0.001 μM (Table 1 and Figure 3b). Also, holo-transferrin had no significant effect on the trypanocidal activity of salinomycin (Table 1: GI_50_ = 0.18 ± 0.06 μM vs. GI_50_ = 0.20 ± 0.02 μM, *p* = 0.2852). These results confirm that ironomycin causes iron depletion in bloodstream-form trypanosomes which can be alleviated by providing additional sources of iron.

Iron depletion induced by ironomycin triggers an increase in the transferrin receptor level in cancer stem cells comparable to the treatment of the cells with the iron chelator deferoxamine [7]. Similarly, bloodstream-form trypanosomes also respond with an upregulation of their transferrin receptor upon iron deprivation induced by deferoxamine [13]. The increase in the transferrin receptor in trypanosomes is associated with an increase in transferrin uptake [13]. It has been previously shown that the upregulation of the trypanosomal transferrin receptor can be usually quantified by measuring the uptake of fluorescently labeled transferrin via flow cytometry [14]. To investigate whether ironomycin treatment leads to an upregulation of the transferrin receptor in bloodstream-form trypanosomes, the parasites were incubated with 250 nM of the ionophore for 19 h followed by quantifying the uptake of fluorescein-labeled transferrin. The uptake of transferrin by trypanosomes exposed to ironomycin was only 61% of that of control cells treated with DMSO (Figure 4a). This result suggests that ironomycin probably induces a downregulation of the transferrin receptor in trypanosomes rather than an upregulation, as observed in cancer stem cells [7]. The downregulation of the transferrin receptor was confirmed by Western blotting. The amount of the transferrin receptor in parasites treated with ironomycin was reduced to 56% compared with control cells (Figure 4b, compare lanes DMSO and IRO). Thus, the level of downregulation of the transferrin receptor is sufficient to explain the reduction in transferrin uptake in trypanosomes exposed to ironomycin. In contrast, treatment of the parasites with 25 μM deferoxamine resulted in a 2.2-fold increase in the uptake of transferrin and a 2.2-fold increase in the transferrin receptor (Figure 4), which is in agreement with previous findings that iron depletion triggers upregulation of the transferrin receptor in trypanosomes [13,14]. Incubation of trypanosomes with 250 nM salinomycin also led to a reduction in the uptake of transferrin, but only by 23% (Figure 4a). However, the amount of the transferrin receptor was increased in salinomycin-treated trypanosomes (Figure 4b, compare lanes DMSO and SAL). This finding suggests that salinomycin may inhibit endocytosis, so less transferrin-loaded transferrin receptor molecules are trafficked to endosomes. A similar observation was made in chronic lymphocytic leukemia cells in which salinomycin blocked the endocytosis of the lipoprotein receptor-related protein 6 [15]. Salinomycin has also been shown to raise the pH of the endosomal-lysosomal system [16], which would mean that transferrin would be less able to release its iron and, thus, not dissociate from the trypanosomal transferrin receptor when trafficking through endocytic compartments. Consequently, less transferrin would be delivered to the lysosome, resulting in less transferrin accumulation in the organelle under the experimental conditions. (It should be noted that, in bloodstream-form trypanosomes, holo-transferrin releases its iron under acidic conditions in the endosome. The iron-free transferrin dissociates from the receptor and is subsequently delivered to the lysosome where it is proteolytically degraded [17].)

Taken together, these findings suggest that the iron deprivation caused by ironomycin in bloodstream-form trypanosomes is due to the downregulation of the transferrin receptor, which leads to reduced uptake of transferrin and, thus, to reduced uptake of iron. The downregulation of the transferrin receptor may be the trypanosomes’ response to reducing the effect of iron-mediated production of ROS indirectly promoted by ironomycin (see below).

### 2.3. Effect of Ironomycin on ROS Production

The anticancer stem cell activity of ironomycin was linked to the production of ROS in lysosomes promoted by sequestered iron(II) in the organelle [7]. To see whether the increased trypanocidal activity of ironomycin is due to the generation of ROS, the effect of antioxidants on the anti-trypanosomal activity of the compound was investigated. First, we tested if the antioxidative tripeptide glutathione (GSH) could reduce the trypanocidal activity of ironomycin. However, the addition of 1 mM GSH reduced only marginally the anti-trypanosomal activity of the ionophore as the GI_50_ value increased only to 0.038 ± 0.002 (Table 1 and Figure 5). This could be because bloodstream-form trypanosomes do not readily take up the peptide [18]. Therefore, we investigated next whether the membrane-permeable ROS scavenger dithiothreitol (DTT) could decrease the anti-trypanosomal activity of ironomycin. As DTT at a concentration of 0.5 mM (this would be equivalent to 1 mM GSH based on sulfhydryl groups present in the molecules: GSH, one SH-group; DTT, two SH-groups) is toxic to bloodstream forms of *T. brucei* [19], we tested 50 μM of the thiol, a concentration that proved to be nontoxic to the parasite (Appendix B, Figure A4). Like GSH, DTT did not reduce the trypanocidal activity of ironomycin (Table 1). Also, the trypanocidal activity of salinomycin was only insignificantly affected by the thiol (Table 1). However, a concentration of 50 μM DTT should have been sufficient to reduce intracellularly generated ROS. For instance, in mouse fibroblasts pretreated with 50 μM DTT, ultraviolet A radiation-induced ROS were reduced by 57% [20]. The finding that the water-soluble antioxidants GSH and DTT were ineffective in reducing the trypanocidal activity of ironomycin may indicate that lysosomal-produced ROS lead to lipid peroxidation of the lysosomal membrane. To assess this possibility, we tested whether the lipophilic antioxidant vitamin E could reduce the trypanocidal activity of ironomycin. This vitamin efficiently inhibits lipid peroxidation caused by ROS in polyunsaturated fatty acids [21]. A highly concentrated preparation containing 70.99% mixed D-tocopherols was used as a source of vitamin E. In the presence of 10 μg/mL vitamin E (23 μM), the trypanocidal activity of ironomycin was substantially reduced, as demonstrated by an increase in the GI_50_ value to 0.18 ± 0.01 μM (Figure 5 and Table 1). Since the trypanocidal activity of salinomycin was also reduced in the presence of vitamin E as indicated by an increase in the GI_50_ value to 0.22 ± 0.02 μM (Figure 5 and Table 1), the difference between the anti-trypanosomal activity of the two ionophores was still statistically significant (0.18 ± 0.01 μM vs. 0.22 ± 0.02 μM, *p* = 0.0147). However, when 50 μM ammonium ferric citrate was also present in addition to 10 μg/mL vitamin E, the trypanocidal activity of ironomycin was further reduced, increasing the GI_50_ value to 0.21 ± 0.02 μM (Figure 5 and Table 1). Under these conditions, the trypanocidal activity of ironomycin was statistically not significantly different from the GI_50_ value of salinomycin in the absence or presence of vitamin E (Table 1: 0.21 ± 0.02 μM vs. 0.20 ± 0.02 μM, *p* = 0.6433; 0.21 ± 0.02 μM vs. 0.22 ± 0.02 μM, *p* = 0.5186). These results suggest that the increased trypanocidal activity of ironomycin can be attributed to lipid peroxidation caused by ROS and iron deprivation.

To confirm that ironomycin triggers the generation of ROS in bloodstream-form trypanosomes, the presence of oxygen-containing free radicals was determined in the parasites after incubation with the ionophore using the cell-permeant ROS indicator 2′,7′-dichlorodihydrofluorescein diacetate (DCFH_2_-DA). Compared with DMSO-treated control cells, the ROS level of trypanosomes incubated with 250 nM ironomycin for 14 h was 3.2 times higher (Figure 6). Also, in trypanosomes treated with 250 nM salinomycin, the ROS production was increased, but only 1.7 times compared with control cells (Figure 6).

These results indicate that bloodstream-form trypanosomes can cope with moderate ROS levels induced by salinomycin but not with higher ROS levels caused by ironomycin. This suggestion is supported by the finding that vitamin E reduced the trypanocidal activity of salinomycin only slightly (from GI_50_ = 0.20 μM to GI_50_ = 0.22 μM, Table 1), whereas the antioxidant lowered the anti-trypanosomal activity of ironomycin considerably (from GI_50_ = 0.034 μM to GI_50_ = 0.18 μM, Table 1). In addition, our findings confirm previous observations in cancer stem cells that ironomycin triggers a more potent ROS production than salinomycin [7].

### 2.4. Effect of Ironomycin on Subcellular Iron(II) Distribution

To determine the impact of ironomycin on the cellular iron distribution in bloodstream-form trypanosomes, the fluorescence probe RhoNox-1 selective for iron(II) was employed. This probe is reduced by iron(II) to the red fluorescent product rhodamine B (Appendix A) [22]. In vehicle-treated control cells, the RhoNox-1 fluorescence signal was diffusely distributed (Figure 4, DMSO), most likely reflecting an even dispersal of iron(II) in the parasite. In trypanosomes treated with ironomycin, a focused fluorescent spot just posterior to the centrally located nucleus was detectable in addition to a weak and diffused fluorescence of the cytoplasm (Figure 7, IRO). The subcellular localization of the fluorescent spot is consistent with the position of the lysosome in bloodstream-form trypanosomes [23]. This result suggests that iron(II) is not transported out of the lysosome but sequestered in the organelle. A similar finding was also made for trypanosomes incubated with salinomycin; however, it is difficult to determine whether the intensity of the signal is different from that observed in cells treated with ironomycin (Figure 7, SAL). Thus, it cannot be confirmed whether salinomycin is less efficient than ironomycin in sequestering iron(II) in the lysosome as the produced ROS levels by the ionophores would suggest (see Figure 6).

### 2.5. Conclusion

This study has shown that the trypanocidal activity of ironomycin comprises three actions: (i) ionophoretic activity, (ii) ROS production, and (iii) iron deprivation. In contrast, the trypanocidal activity of its parent compound salinomycin is mainly due to ionophoretic activity. Previously it was demonstrated that in cancer stem cells, ironomycin accumulates in the lysosome where it sequesters iron(II) leading to cytoplasmic iron depletion [7]. The lysosomal-sequestered iron catalyzes the production of ROS via Fenton chemistry [24]. Likewise, ironomycin works the same way in bloodstream-form trypanosomes, i.e., by trapping iron(II) in the lysosome where the metal mediates the production of ROS. The consequence of the iron(II) sequestration in the lysosome is that the metal is not released from the organelle into the cytoplasm leading to cellular iron deprivation. Usually, iron depletion results in an upregulation of transferrin uptake in bloodstream-form trypanosomes [13,14]. However, in the case of ironomycin-mediated cytoplasmic iron deprivation, the parasite’s response is a downregulation of transferrin uptake. With this response, bloodstream-form trypanosomes could reduce iron uptake to limit the damage caused by ROS produced by lysosomal-trapped iron(II). Ironomycin’s parent compound salinomycin also triggers the production of ROS and causes iron deprivation in bloodstream-form trypanosomes, but to a much lesser extent. As iron supplementation and vitamin E did not affect or only marginally affected the trypanocidal activity of salinomycin, it seems the parasite can cope with the moderate iron deficiency and ROS levels induced by the ionophore. Hence, salinomycin derivatives that effectively sequester iron(II) in lysosomes are a valuable strategy for anti-trypanosomal drug development.

## 3. Materials and Methods

### 3.1. Reagents

Ammonium ferric citrate (09713, Fluka/Fisher Scientific, Loughborough, UK), Benzyloxycarbonyl-phenylalanyl-alanyl-diazomethyl ketone (4003801, Bachem, Budendorf, Switzerland), bovine apo-transferrin (T1428, Sigma-Aldrich/Merck, Darmstadt, Germany), Bovine holo-transferrin (T1283, Sigma-Aldrich/Merck, Darmstadt, Germany), bovine serum albumin (BSA, A3059, Sigma-Aldrich/Merch, Darmstadt, Germany), ReadyBlue™ protein gel stain (RSB-1L, Sigma-Aldrich/Merck, Darmstadt, Germany), deferoxamine (D9533, Sigma-Aldrich/Merck, Darmstadt, Germany), 2′,7′-dichlorodihydrofluorescein diacetate (DCFH_2_-DA, 35845, Sigma-Aldrich/Merck, Darmstadt, Germany), IRDye^®^ 800 CW donkey anti-rabbit IgG (926-32213, LICORbio, Lincoln, NE, USA), 6-[Fluorescein-5(6)-carboxamido]hexanoic acid *N*-hydroxysuccinimide ester (46940, Sigma-Aldrich/Merck, Darmstadt, Germany), glutathione (G4251, Sigma-Aldrich/Merck, Darmstadt, Germany), high-strength Vitamin E 70% in sunflower carrier oil (No. 827, Naissance, Neath, UK), human holo-transferrin (T0665, Sigma-Aldrich/Merck, Darmstadt, Germany).

### 3.2. Ionophores

Salinomycin was purified from the veterinary feed additive Sacox^®^ as previously described [25]. In brief, the feed additive was dissolved in dichloromethane. Crude product was obtained by evaporating the solvent under reduced pressure. The crude product was purified by dry-column vacuum chromatography (gradient solvent mixture dichloromethane/acetone) providing the sodium salt of salinomycin. To obtain the free acid of the ionophore, the sodium salt of salinomycin was dissolved in dichloromethane and vigorously extracted with aqueous sulphuric acid (pH 1.0). The organic phase was washed with distilled water and the solvent evaporated under reduced pressure. After three further evaporation steps, salinomycin was obtained as a white powder. The spectroscopic data of salinomycin were in agreement with previously published data [25].

Ironomycin was synthesized following the procedure detailed by Mai et al. (2017) [7]. The spectroscopic data of ironomycin were in line with previously published data [5]: ^1^H NMR (403 MHz, CD_2_Cl_2_) δ 6.20 (dd, *J* = 10.4, 1.2 Hz, 1H), 6.06 (dd, *J* = 10.3, 4.3 Hz, 1H), 4.17 (dd, *J* = 10.1, 1.5 Hz, 1H), 3.97 (dd, *J* = 16.2, 2.5 Hz, 1H), 3.90 (dd, *J* = 10.9, 5.5 Hz, 1H), 3.86 (dd, *J* = 16.2, 2.5 Hz, 1H), 3.75 (dd, *J* = 13.6, 6.8 Hz, 1H), 3.65 (dd, *J* = 8.2, 6.0 Hz, 2H), 3.60–3.55 (m, 2H), 2.87 (td, *J* = 10.8, 4.1 Hz, 1H), 2.65–2.54 (m, 2H), 2.28 (t, *J* = 2.5 Hz, 1H), 2.13–1.10 (m, 34H), 0.98–0.85 (m, 12H), 0.85–0.71 (m, 9H), 0.67 (d, *J* = 6.9 Hz, 3H) ppm; ^13^C NMR (101 MHz, CD_2_Cl_2_) δ 214.3, 178.4, 128.8, 127.0, 107.1, 98.9, 88.7, 80.4, 77.5, 75.63, 75.56, 73.7, 73.3, 71.9, 71.2, 68.4, 55.8, 50.3, 48.6, 40.2, 39.7, 38.6, 36.7, 36.4, 32.9, 31.5, 30.9, 30.1, 29.4, 28.4, 26.8, 25.2, 23.6, 22.2, 20.2, 17.7, 16.5, 15.9, 14.5, 13.2, 12.8, 12.2, 11.2, 7.1, 6.5 ppm; HRMS (ESI^+^) *m*/*z* [M+H]^+^ Calcd for C_45_H_74_NO_10_^+^ 788.5307, Found 788.5299.

### 3.3. Synthesis of RhoNox-1

The fluorescence iron(II) probe RhoNox-1 was synthesized following the protocol published by Hirayama et al. (2013) [22]. The spectroscopic data of RhoNox-1 are in line with those found in the reference literature (for ^1^H-NMR and ^13^C-NMR spectra see Appendix A).

### 3.4. Labelling of Transferrin with Fluorescein

Human holo-transferrin was labeled at a molar ratio of 1:22 with 6-[fluorescein-5(6)-carboxamido]hexanoic acid *N*-hydroxysuccinimide ester [26]. The unreacted labeling reagent was removed using a PD-10 column equilibrated with PBS. Eluate containing fluorescein-labeled transferrin was collected and stored at −20 °C.

### 3.5. Culturing of Trypanosomes

The monomorphic 427-221a clone of bloodstream forms of *T. brucei* [27] was cultured in Baltz medium [28] supplemented with 16.7% heat-inactivated bovine serum. The cultures were grown in a humidified incubator with 5% carbon dioxide at 37 °C.

### 3.6. Cell Swelling Assay

Cell volume changes in flagellates can be determined by the light scattering technique monitoring the absorbance of cell suspensions at 450–550 nm whereby a decrease in absorbance indicates swelling of cells [29]. Based on the filter availability of the BioTek ELx808 microplate reader (Fisher Scientific, UK), cell swelling of trypanosomes was measured at 490 nm as previously described [6]. In brief, bloodstream-form trypanosomes were incubated at a density of 5 × 10^7^ cells/mL in 96-well plates in a final volume of 200 µL Baltz medium containing 100 µM ionophore and 0.9% DMSO. The absorbance of the cultures was measured every 10 min for 1 h. At the end of the experiment, the trypanosomes were checked for motility as an indicator of vitality.

### 3.7. In Vitro Toxicity Assay

The trypanocidal activity of ionophores was evaluated according to previously published protocols [4,5]. Bloodstream forms of *T. brucei* 427-221a were seeded into wells of a 96-well plate at an initial cell density of 1 × 10^4^/mL in 200 μL of Baltz medium supplemented with 16.7% heat-inactivated bovine serum in the absence or presence of ammonium ferric citrate, transferrin, and/or antioxidants. The ionophores were assayed at tenfold dilutions, beginning with 100 μM down to 100 nM in the presence of 0.9% DMSO. Wells containing medium with 0.9% DMSO in the absence or presence of ammonium ferric citrate, transferrin, and/or antioxidants served as controls. After incubation of the cultures for 24 h, 20 μL of a 0.5 mM resazurin solution in PBS was added to each well. After a further 48 h of incubation, the absorbance of each well at 570 nm (test wavelength) and 630 nm (reference wavelength) was read on a BioTek ELx808 microplate reader (Fisher Scientific, UK). The half-maximal growth inhibitory (GI_50_) value, i.e., the concentration of an ionophore necessary to reduce the growth rate of trypanosomes by 50% compared with the control, was determined by linear interpolation as described by Huber and Koella (1993) [30].

### 3.8. Transferrin Uptake Measurement

Uptake of fluorescein-labeled transferrin was performed as previously described [31]. Bloodstream forms of *T. brucei* 427-221a were incubated at a cell density of 5 × 10^5^/mL with 250 nM ironomycin or salinomycin in the presence of 1% DMSO in Baltz medium supplemented with 16.7% bovine serum at 37 °C for 19 h in a CO_2_-incubator. Controls were treated either with 1% DMSO or with 25 μM deferoxamine. At the end of the incubation period, cells were harvested and washed once with 10 mL and twice with 5 mL Baltz medium supplemented with 2% BSA by centrifugation at 1620× *g* for 10 min. Then, the trypanosomes (6–8 × 10^6^/mL) were incubated with 100 μg/mL fluorescein-labeled human holo-transferrin in Baltz medium supplemented with 2% BSA in the presence of 100 μM benzyloxycarbonyl-phenylalanyl-alanyl-diazomethyl ketone (to prevent the degradation of transferrin) for 2 h. Subsequently, the cells were harvested and washed twice with 1 mL PBS/1% glucose by centrifugation at 755× *g* for 5 min. After fixing the cells in 2% formaldehyde/0.05% glutaraldehyde in PBS, accumulated transferrin within the trypanosomes was determined by flow cytometry using a BD FACSymphony cell analyzer (Wokingham, UK).

### 3.9. Transferrin Receptor Level Determination

The expression of the transferrin receptor in the bloodstream forms of *T. brucei* was determined by Western blotting. To this end, trypanosomes were first incubated with ironomycin (250 nM), salinomycin (250 nM), DMSO (1%), or deferoxamine (25 μM), as described in Section 3.8. After incubation, cells were harvested by centrifugation at 1620× *g* for 10 min and washed thrice with 1 mL PBS/1% glucose by centrifugation at 755× *g* for 5 min. Then, the trypanosomes were lysed in an SDS/PAGE sample buffer at 5 × 10^5^ cells/20 μL. Parasite lysates (5 × 10^5^ cells) were analyzed by SDS-PAGE on 4–15% precast polyacrylamide gels (Bio-Rad, Watford, UK) and stained with ReadyBlue™ protein gel stain (to confirm equal protein amounts in the samples) or electroblotted onto a PVDF membrane. The blot was briefly washed three times with TBST (50 mM Tris, 150 mM NaCl, pH 7.5, 0.05% Tween 20), blocked with 5% milk powder in TBST for 30 min, and then incubated with purified rabbit anti-*T. brucei* transferrin receptor IgG (anti-rabbit TFBP IgG [24]) diluted 1:200 in TBST/5% milk powder overnight at 4 °C. Subsequently, the blot was washed four times with TBST for 15 min and then incubated with IRDye^®^ 800 CW donkey anti-rabbit IgG dilute 1:10,000 in TBST/5% milk powder for 2 h at room temperature. After four more washes with TBST for 15 min, the blot was imaged using the LI-COR Odyssey CLX imaging system (Lincoln, NE, USA).

### 3.10. Determination of ROS Production

Intracellular ROS levels were measured using the cell-permeant ROS indicator 2′,7′-dichlorodihydrofluorescein diacetate (DCFH_2_-DA). This probe has been previously used to determine ROS generation in *T. evansi* [32], a closely related trypanosome species to *T. brucei*. DCFH_2_-DA can easily enter cells by passively crossing cell membranes [33]. Within cells, the probe is cleaved by intracellular esterases to form 2′,7′-dichlorodihydrofluorescein (DCFH_2_), which is also non-fluorescent but membrane-impermeable and, therefore, retained within cells [33]. DCFH_2_ reacts with ROS to form the fluorescent compound 2′,7′ -dichlorofluorescein (DCF) [33]. Bloodstream forms of *T. brucei* 427-221a were washed twice with 10 mL PBS/1% glucose by centrifugation at 1620× *g* for 10 min and then pre-loaded with 50 μM DCFH_2_-DA at a cell density of 2 × 10^7^/mL in PBS/1% glucose at 37 °C for 30 min. Subsequently, the trypanosomes were incubated at a cell density of 5 × 10^5^/mL with 250 nM ironomycin or salinomycin in the presence of 1% DMSO in Baltz medium supplemented with 16.7% bovine serum at 37 °C in a CO_2_-incubator. After 14 h incubation, the cells were harvested and washed twice with 1 mL PBS/1% glucose by centrifugation at 755× *g* for 5 min. After fixing the cells in 2% formaldehyde/0.05% glutaraldehyde in PBS, the fluorescence of the cells was measured by flow cytometry using a BD FACSymphony cell analyzer (Wokingham, UK). The percentage of ROS produced within trypanosomes was calculated according to the equation
(1)$$\%ROS=\frac{{FL}_{sample}-{FL}_{auto}}{{FL}_{auto}}\times 100$$
with FL_sample_ and FL_auto_ being the mean fluorescence intensity signal of the test sample and the autofluorescence of unstained cells, respectively.

### 3.11. Localization of Intracellular Iron(II)

Bloodstream forms of *T. brucei* were incubated with 250 nM ironomycin or salinomycin, or DMSO (1%) as described in Section 3.8. Cells were harvested by centrifugation at 1620× *g* for 10 min and washed twice with 1 mL PBS/1% glucose by centrifugation at 755× *g* for 5 min. The trypanosomes were then incubated with 10 μM RhoNox-1 in PBS/1% glucose at a cell density of 4 × 10^6^/mL for 30 min at 37 °C in a CO_2_-incubator. Subsequently, cells were harvested and washed once with 1 mL PBS/1% glucose by centrifugation at 755× *g* for 5 min, and fixed in 2% formaldehyde/0.05% glutaraldehyde in PBS. The fixed cells were applied to poly-L-lysine-coated microscope slides and stained with 0.0001% DAPI in PBS. Slides were covered in Vectashield (2BScientific Ltd., Kidlington, UK) mounting medium and examined with a Zeiss Apotome Axio Imager.M2 microscope (Cambourne, UK) using a 63× oil immersion objective.

## Figures and Tables

**Figure 1 molecules-29-05597-f001:**
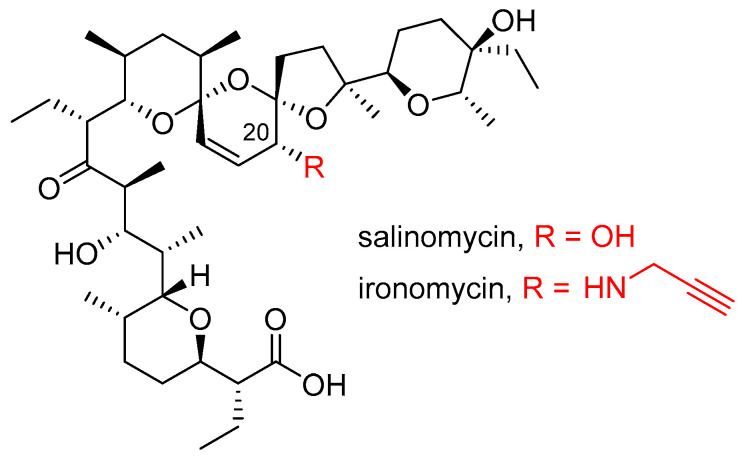
Chemical structure of salinomycin and ironomycin. The PubChem compound identifier (CID) is 3085092 and 155520759, respectively.

**Figure 2 molecules-29-05597-f002:**
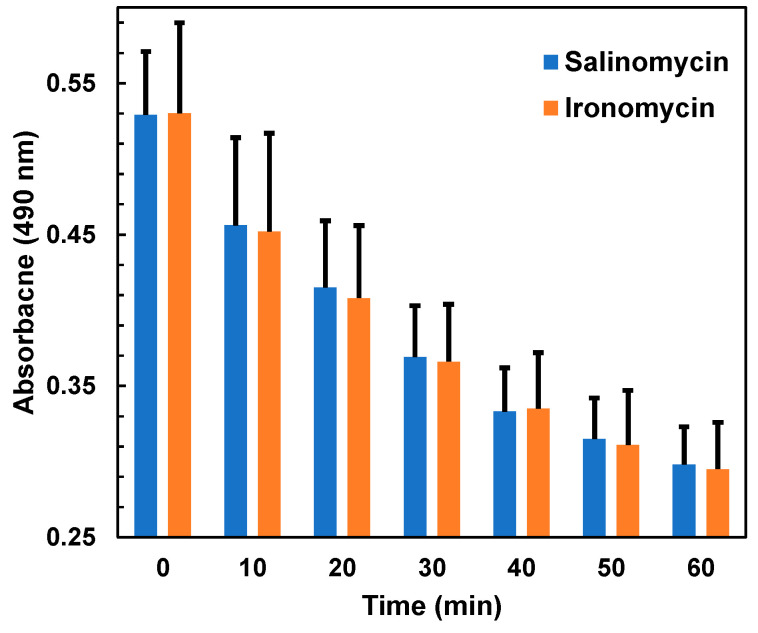
Effect of salinomycin and ironomycin on the cell volume of bloodstream forms of *T. brucei*. The parasites were incubated at a cell density of 5 × 10^7^/mL with 100 μM ionophore in the presence of 0.9% DMSO. Every 10 min, the absorbance at 490 nm was read. It should be noted that a decrease in absorbance corresponds to an increase in cell volume. No significant change in cell volume was observed for trypanosomes incubated with only 0.9% DMSO during the measurement period (see Appendix B, Figure A1). Mean values ± SD of three experiments are shown.

**Figure 3 molecules-29-05597-f003:**
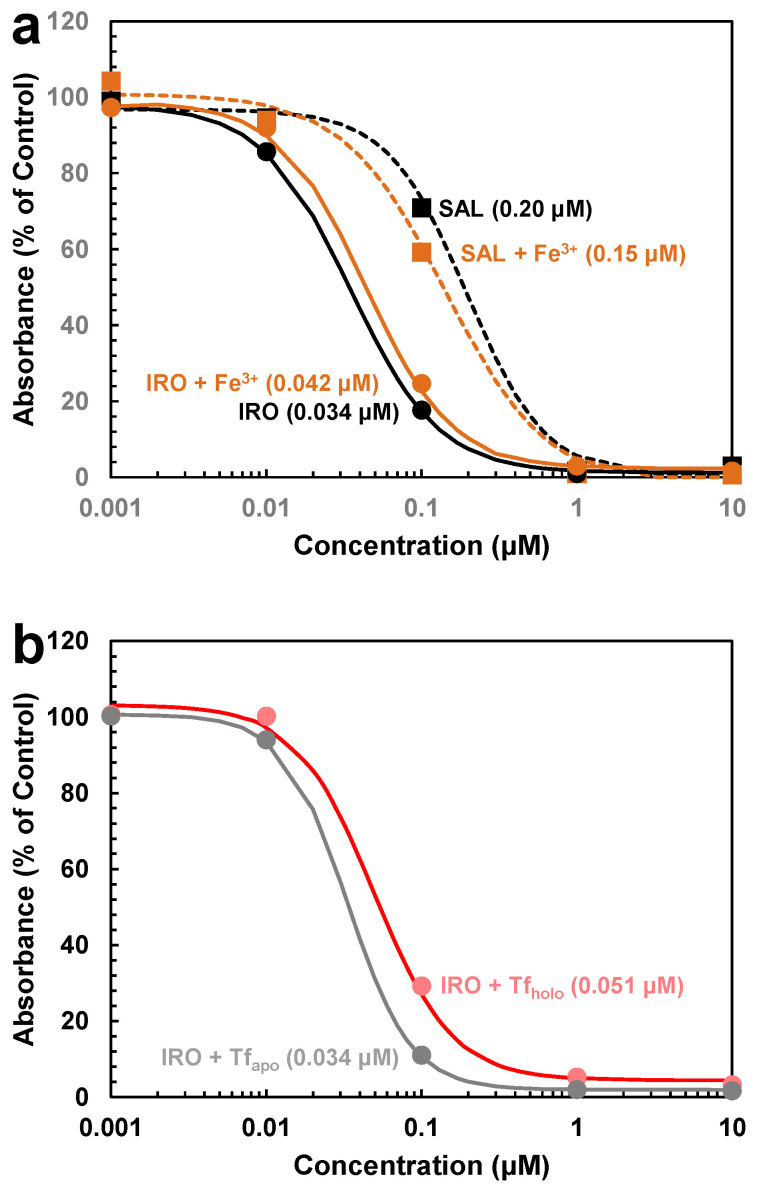
Effect of iron on the trypanocidal activity of ironomycin and salinomycin. Bloodstream forms of *T. brucei* were incubated (**a**) with varying concentrations of ironomycin (IRO, circles, solid lines) or salinomycin (SAL, squares, dashed lines) in the absence (black symbols and lines) or presence (ochre symbols and lines) of 50 μM ammonium ferric citrate (Fe^3+^) or (**b**) with varying concentration of ironomycin (IRO, circles, solid lines) in the presence of 1.2 mg/mL bovine apo-transferrin (Tf_apo_ (iron-free Tf), grey symbols and line) or 1.2 mg/mL bovine holo-transferrin (Tf_holo_ (iron-saturated Tf), pink symbols and line). After 72 h of culture, the viability and proliferation of the cells were determined with the colorimetric dye resazurin. The mean values of three independent experiments are shown. For clarity, standard deviations were omitted. The standard deviations ranged from 0.6 to 11.0 percentage points. The dose–response curves were calculated based on the mean values using the 4-parameter logistic regression model. Numbers in brackets represent GI_50_ values.

**Figure 4 molecules-29-05597-f004:**
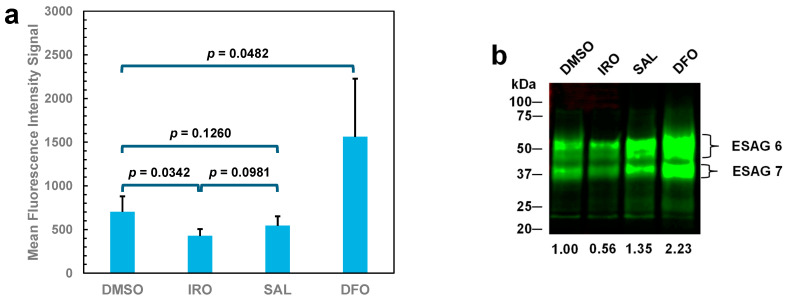
Accumulation of fluorescein-labeled transferrin and expression of the transferrin receptor in bloodstream forms of *T. brucei* upon incubation with ionophores and deferoxamine. Trypanosomes were incubated first with 250 nM ironomycin (IRO) or salinomycin (SAL), with 25 μM deferoxamine (DFO), or only with 1% DMSO (control) for 19 h. (**a**) Trypanosomes were subsequently incubated with 100 μg/mL fluorescein-labeled human transferrin in the presence of 100 μM of the cysteine protease inhibitor benzyloxycarbonyl-phenylalanyl-alanyl-diazomethyl ketone for 2 h and then analyzed by flow cytometry. The mean values and standard deviations of the three experiments are shown. (**b**) Detection of the transferrin receptor (expression site-associated gene (ESAG) 6/7 heterodimer) in lysates of trypanosomes (5 × 10^5^ cells per lane) by Western blotting using anti-*T. brucei* transferrin receptor antibodies. Numbers below the blot represent the relative band intensities (DMSO control was set to 1) normalized based on the intensity of the variant surface glycoprotein bands in the samples after Coomassie Blue staining (see Appendix B, Figure A3).

**Figure 5 molecules-29-05597-f005:**
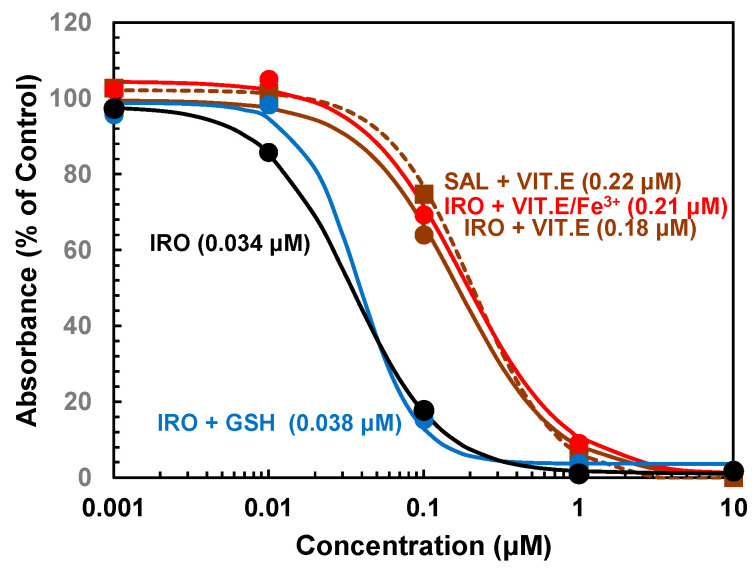
Effect of antioxidants on the trypanocidal activity of ironomycin and salinomycin. Bloodstream forms of *T. brucei* were incubated with varying concentrations of ironomycin (IRO, circles, solid lines) or salinomycin (SAL, squares, dashed lines) in the absence (black symbols and line) or in the presence of 1 mM glutathione (GSH, blue symbols and line), 10 μg/mL vitamin E (VIT.E, brown symbols and lines), or 10 μg/mL vitamin E plus 50 μM ammonium ferric citrate (VIT.E/Fe^3+^, red-orange symbols and line). After 72 h of culture, the viability and proliferation of the cells were determined with the colorimetric dye resazurin. The mean values of three independent experiments are shown. For clarity, standard deviations were omitted. The standard deviations ranged from 1.7 to 7.1 percentage points. The dose–response curves were calculated based on the mean values using the 4-parameter logistic regression model. Numbers in brackets represent GI_50_ values.

**Figure 6 molecules-29-05597-f006:**
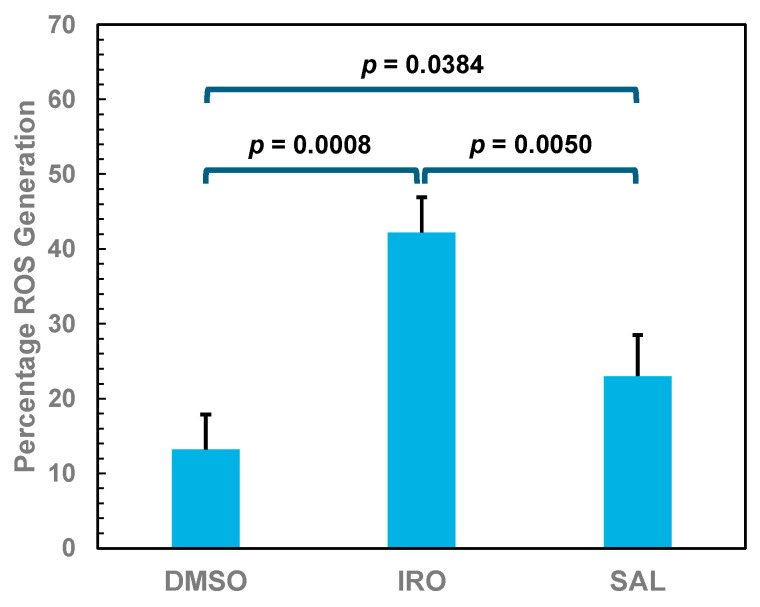
Percentage of ROS generation in bloodstream forms of *T. brucei* upon incubation with ionophores. Trypanosomes were first loaded with DCFH_2_-DA for 30 min, followed by incubation with 250 nM ironomycin (IRO) or salinomycin (SAL), or only with 1% DMSO (control). After 14 h incubation, trypanosomes were analyzed by flow cytometry. The mean values and standard deviations of the three experiments are shown.

**Figure 7 molecules-29-05597-f007:**
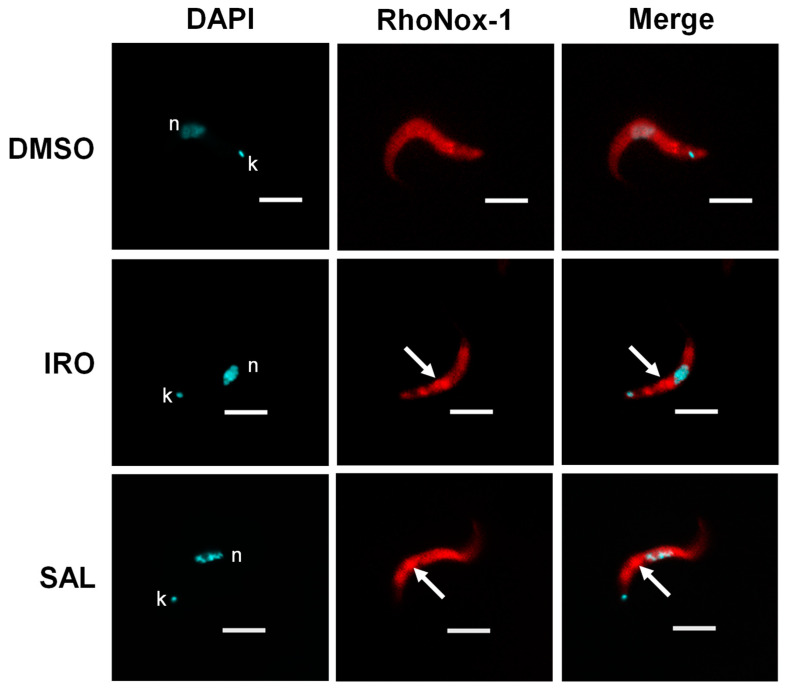
Distribution of iron(II) in bloodstream forms of *T. brucei* upon incubation with ionophores. Trypanosomes were treated first with 250 nM ironomycin (IRO) or salinomycin (SAL) or only with 1% DMSO (control) for 19 h. Then the cells were incubated with 10 μM RhoNox-1 for 30 min and inspected by fluorescence microscopy. Red fluorescent spots indicated by arrows represent sequestered iron(II) in lysosomes located between the nucleus (n) and kinetoplast (k). Bar = 5 μm.

**Table 1 molecules-29-05597-t001:** GI_50_ values of ironomycin and salinomycin against bloodstream forms of *T. brucei* under different experimental conditions.

Experimental Condition	GI_50_ ± SD ^a^ (μM)	
	Ironomycin	Salinomycin
No addition	0.034 ± 0.002	0.20 ± 0.02
Ammonium ferric citrate (50 μM)	0.042 ± 0.002	0.15 ± 0.02
Holo-transferrin (1.2 mg/mL)	0.051 ± 0.005	0.18 ± 0.06
Apo-transferrin (1.2 mg/mL)	0.034 ± 0.001	n.d. ^b^
Glutathione (GSH) (1 mM)	0.038 ± 0.002	n.d.
Dithiothreitol (DTT) (50 μM)	0.032 ± 0.006	0.22 ± 0.03
Vitamin E (10 μg/mL)	0.18 ± 0.01	0.22 ± 0.02
Vitamin E (10 μg/mL) + ammonium ferric citrate (50 μM)	0.21 ± 0.02	n.d.

^a^: mean values and standard deviation of three independent experiments. ^b^: not determined.

## Data Availability

Data are contained within the article.

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
