# Peer review of "Increased Trypanocidal Activity of the Salinomycin Derivative Ironomycin Is Due to ROS Production and Iron Uptake Impairment"

_molecules, 2024, doi:10.3390/molecules29235597_

Round 1
Reviewer 1 Report
Comments and Suggestions for Authors
The work entitled “Increased trypanocidal activity of the salinomycin derivative ironomycin is due to ROS production and iron uptake impairment” by Steverding and collaborators seeks to determine the causes of the increase in trypanocidal activity of ironomycin.
In general terms, the manuscript is very well written, the objectives, results and discussion are written consistently and clearly.
The introduction presents an appropriate wording, although, considering that African trypanosomiasis is considered by the WHO as a neglected tropical disease, it is important that the authors briefly relate some epidemiological aspects of the disease, in order to highlight the importance of the work done.
Considering the above and with the minimum suggestions, the manuscript is suitable for publication.
Author Response
The work entitled “Increased trypanocidal activity of the salinomycin derivative ironomycin is due to ROS production and iron uptake impairment” by Steverding and collaborators seeks to determine the causes of the increase in trypanocidal activity of ironomycin.
In general terms, the manuscript is very well written, the objectives, results and discussion are written consistently and clearly. Reply: We thank the Reviewer for their time in critically reading the manuscript and the positive assessment of our work.
The introduction presents an appropriate wording, although, considering that African trypanosomiasis is considered by the WHO as a neglected tropical disease, it is important that the authors briefly relate some epidemiological aspects of the disease, in order to highlight the importance of the work done. Reply: A new first paragraph describing the causative agents and the epidemiology of human and animal African trypanosomiasis has been included in the Introduction (see pages 1-2, lines 37-48).
Considering the above and with the minimum suggestions, the manuscript is suitable for publication. Reply: We hope that we have adequately addressed the reviewer’s recommendations.
Reviewer 2 Report
Comments and Suggestions for Authors
In the manuscript: "Increased trypanocidal activity of the salinomycin derivative ironomycin is due to ROS production and iron uptake impairment", the authors aimed to elucidate the mechanisms behind the trypanocidal effects of salinomycin and ironomycin on bloodstream-form Trypanosoma brucei. They concentrated on the ionophoretic activity of these compounds, which leads to sodium cation influx and osmotic water absorption, their ROS production capability, and their impact on iron uptake.
Nevertheless, despite the authors' efforts, the manuscript does not exhibit the originality needed for publication in this journal. The initial results are simply a validation of earlier findings from the research group, and the potentially novel results not only conflict with existing data from other models but are also inconsistent internally.
The authors are advised to conduct additional experiments to support their conclusions, such as demonstrating Fe accumulation in lysosomes.
Moreover, the authors should: 1) include standard deviations in graphs 3 and 5; 2) indicate statistical significance, if applicable, in graphs 4 and 6; 3) report the initial Fe concentration in the medium (given that it was supplemented with bovine serum); 4) minimize self-citations; 5) update the bibliography to enhance the discussion of their results.
If the authors implement these suggestions, the manuscript could enhance the understanding of how these drugs exert their trypanocidal effects and their potential use in treating parasitic diseases.
Author Response
In the manuscript: "Increased trypanocidal activity of the salinomycin derivative ironomycin is due to ROS production and iron uptake impairment", the authors aimed to elucidate the mechanisms behind the trypanocidal effects of salinomycin and ironomycin on bloodstream-form Trypanosoma brucei. They concentrated on the ionophoretic activity of these compounds, which leads to sodium cation influx and osmotic water absorption, their ROS production capability, and their impact on iron uptake.
Nevertheless, despite the authors' efforts, the manuscript does not exhibit the originality needed for publication in this journal. The initial results are simply a validation of earlier findings from the research group, and the potentially novel results not only conflict with existing data from other models but are also inconsistent internally. Reply: We thank the Reviewer for their time in critically reading the manuscript and their valuable comments.
The authors are advised to conduct additional experiments to support their conclusions, such as demonstrating Fe accumulation in lysosomes. Reply: We have determined the subcellular distribution of iron(II) in bloodstream-form trypanosomes using the fluorescence probe RhoNox-1 (see page 12, lines 464-474). To be able to do this, we had to synthesize RhoNox-1 (see page 10, lines 371-375 and Supplementary Materials, Figure S1). The results of this experiment are described in a new paragraph (see pages 8-9, lines 292-307) and are shown in a new figure (see page 9, Figure 7, and lines 309-314). It was found that both ionophores lead to an accumulation of iron(II) in the parasite’s lysosome.
Moreover, the authors should:
1) include standard deviations in graphs 3 and 5. Reply: We have purposely omitted the standard deviations in Figures 3 and 5 for clarity reasons. If standard deviations were included in these figures, the course of the curves would be obstructed, especially in cases where data points are close together. This was mentioned in the figure legends.
2) indicate statistical significance, if applicable, in graphs 4 and 6. Reply: As suggested by the reviewer, statistical significance has been included in Figures 4 and 6 (see Figure 4a on page 6 and Figure 6 on page 8).
3) report the initial Fe concentration in the medium (given that it was supplemented with bovine serum). Reply: The amount of free iron in the culture medium is extremely low. This is because transferrin supplemented with the bovine serum restricts the concentration of free iron to 10-18 M (Bullen (1981) Rev. Infect. Dis. 3, 1127-38). This is now stated in the manuscript (see page 4, lines 132-134).
4) minimize self-citations. Reply: The old references 3, 4, 5, 8, and 16 have been removed.
5) update the bibliography to enhance the discussion of their results. Reply: 13 new references have been added (see references 1, 2, 3, 8, 10, 15, 16, 17, 19, 20, 21, 22, and 23).
If the authors implement these suggestions, the manuscript could enhance the understanding of how these drugs exert their trypanocidal effects and their potential use in treating parasitic diseases. Reply: We hope that we have adequately addressed the comments.
Reviewer 3 Report
Comments and Suggestions for Authors
Steverding et al. extended the investigation about the mode of action of two related and ionophoric compounds, namely salinomycin and ironomycin, on bloodstream Trypanosoma brucei brucei. Similar approaches to those reported in this manuscript have been used to address this question in different mammalian cells. Thus, the study's novelty lies in the pathogenic organism selected for this research, which causes animal African trypanosomiasis and is closely related to species affecting humans.
The research and results obtained support, to a large extent, the conclusions. Furthermore, the manuscript is well-written.
I do have the following suggestions to be considered by the authors:
Major point
- due to the hydrophobic nature of Vitamin E (or D-tocopherols), it will mainly scavenge ROS generated in unsaturated fatty acids. This should be commented.
- knowing that glutathione does not efficiently cross (or is transported across) Trypanosoma membranes, why not use dithiothreitol as a water-soluble and membrane permeant scavenger of ROS.
- based on the results obtained, it is stated that ironomycin reduces iron uptake in T. brucei by downregulating the expression of the transferrin receptor (TfR). I do not fully agree with this statement because the experiment performed does not provide direct evidence about this, it only informs about a deficiency in the uptake process; which may have different origins. To support this statement, the authors should demonstrate that the expression of the TfR is silenced, eg. by W. blot.
Furthermore, the authors should take into account that a large fraction of the trypanosomal transferrin receptor is rapidly recycled from post-endolysosomal vesicles (i.e. upon release of transferrin in the lysosomal compartment). This has been demonstrated by Kabiri & Steverding, 2000 (DOI: 10.1046/j.1432-1327.2000.01361.x) and is reviewed in Manta et al. 2012 (DOI: 10.5772/34402). Thus, I recommend the authors consider the following hypothetical scenario: accumulation of ironomycin in the lysosome triggers ferroptosis, which impairs the functions of this organelle, one of them being the release of Tf from its receptor and the efficient recycling of the TfR to the parasite membrane. Under such a scenario, the capacity of the parasite to take up extracellular iron bound to Tf would be heavily affected.
- provide some information about the chemistry behind the capacity of these molecules to promote iron accumulation in the lysosome. Are they iron chelators? or simply affect the function of organelles where iron is trafficked?
- comment on the pKa of salinomycin and ironomycin. Are they lysosomotropic agents? Ironomycin has a secondary amine with a pKa close to 10, and hence, might readily accumulate in the lysosome, but is difficult to infer this for salinomycin. Please comment on this.
Minor points
-at several points in the text absolute GI50 values (with standard errors) obtained under different experimental conditions (Fe replenishment, transferrin, Vit E) are compared for both compounds. A Table summarizing these results may help to compare the full set of data. In some cases, the comparisons can be made using relative comparisons (i.e. fold or % increase/decrease of compound potenty).
- the methodological details in the figures' captions are a bit too long and redundant with those described in the Mat&Met section. Thus, it would be convenient to shorten the captions.
- Caption Fig. 2: include "not shown" for the condition treated with only 0.9% DMSO.
- line 78: remove one dash from "...C20--propargylamine...."
- delete "in Bloodstream-Form Trypanosomes" in all subheadings, because at several points of the manuscript, it is stated that the results presented are restricted to performed on this stage and organism.
- line 287: indicate whether the T. b. brucei cell line used corresponds to a strain with capacity (pleomorphic) or not (monomorphic) to complete the full differentiation cycle of the parasite.
- line 325 or 163: indicate why a protease inhibitor was included in the assay (I guess to avoid transferrin degradation).
- line 365-366: indicate in % the decrease in viability excerted by 50 uM ferric iron.
Author Response
Steverding et al. extended the investigation about the mode of action of two related and ionophoric compounds, namely salinomycin and ironomycin, on bloodstream Trypanosoma brucei brucei. Similar approaches to those reported in this manuscript have been used to address this question in different mammalian cells. Thus, the study's novelty lies in the pathogenic organism selected for this research, which causes animal African trypanosomiasis and is closely related to species affecting humans.
The research and results obtained support, to a large extent, the conclusions. Furthermore, the manuscript is well-written. We thank the Reviewer for their time in critically reading the manuscript and their useful comments.
I do have the following suggestions to be considered by the authors:
Major point
- due to the hydrophobic nature of Vitamin E (or D-tocopherols), it will mainly scavenge ROS generated in unsaturated fatty acids. This should be commented. Reply: It is now mentioned in the manuscript that vitamin E efficiently inhibits lipid peroxidation caused by ROS in polyunsaturated fatty acids (see page 7, lines 243-244, and 258-259).
- knowing that glutathione does not efficiently cross (or is transported across) Trypanosoma membranes, why not use dithiothreitol as a water-soluble and membrane permeant scavenger of ROS. Reply: We did not use DTT as the thiol was reported to be toxic to bloodstream forms of T. brucei, at least at concentrations ≥0.5 mM (see Tiengwe et al. (2015) Eukaryotic Cell 14, 1094-1101). Note that 0.5 mM DTT is equivalent to 1 mM GSH (the concentration of GSH we used in the assay) based on sulfuryl groups present in the thiols: DTT, 2 SH-groups; GSH, 1 SH-group. Nevertheless, we investigated whether lower concentrations of DTT are nontoxic to the parasite and found that 50 μM of the thiol does not affect the growth of the trypanosomes whereas 500 μM killed all trypanosomes within 24 h of incubation (see page 14, Figure A4, and lines 510-514). Based on this finding, we tested whether 50 μM DTT could reduce the trypanocidal activity of ironomycin. However, like GSH, DTT did not reduce the trypanocidal activity of ironomycin. These findings are now described in the manuscript (see page 7, lines 229-242).
- based on the results obtained, it is stated that ironomycin reduces iron uptake in T. brucei by downregulating the expression of the transferrin receptor (TfR). I do not fully agree with this statement because the experiment performed does not provide direct evidence about this, it only informs about a deficiency in the uptake process; which may have different origins. To support this statement, the authors should demonstrate that the expression of the TfR is silenced, eg. by W. blot. Reply: As suggested by the reviewer, we determined the expression level of the transferrin receptor in ionophore-treated bloodstream-form trypanosomes by Western blotting The results are presented in the new Figure 4b and are described in the manuscript (see page 5, lines 177-181, and 183-184; page 6, Figure 4b, and lines 214-219; and page 11-12, lines 425-442; and page 14, Figure A3, and lines 503-508). In the case of ironomycin, the amount of the transferrin receptor in the parasites is reduced to a similar extent as the uptake of transferrin. However, in the case of salinomycin, the amount of the transferrin receptor is increased 1.36-fold. We have provided explanations for why salinomycin treatment may result in reduced transferrin uptake despite increased levels of transferrin receptor (see pages 5-6, lines 187-200).
Furthermore, the authors should take into account that a large fraction of the trypanosomal transferrin receptor is rapidly recycled from post-endolysosomal vesicles (i.e. upon release of transferrin in the lysosomal compartment). This has been demonstrated by Kabiri & Steverding, 2000 (DOI: 10.1046/j.1432-1327.2000.01361.x) and is reviewed in Manta et al. 2012 (DOI: 10.5772/34402). Thus, I recommend the authors consider the following hypothetical scenario: accumulation of ironomycin in the lysosome triggers ferroptosis, which impairs the functions of this organelle, one of them being the release of Tf from its receptor and the efficient recycling of the TfR to the parasite membrane. Under such a scenario, the capacity of the parasite to take up extracellular iron bound to Tf would be heavily affected. Reply: The proposed scenario is not possible because it is based on wrong assumptions regarding the iron uptake mechanism in bloodstream forms of T. brucei. After iron-loaded transferrin (holo-transferrin) is bound by the transferrin receptor in the flagellar pocket, the receptor-ligand-complex is endocytosed and trafficked to the endosome. The low pH of the endosome triggers the release of iron from transferrin. Iron-free transferrin (apo-transferrin) then dissociates from the receptor in the endosome. While the transferrin receptor is recycled to the flagellar pocket to bind new holo-transferrin, apo-transferrin is delivered to the lysosome where it is proteolytically degraded (see Steverding 2000. Parasitol. Int. 48, 191-198). As mentioned above, we have provided alternative explanations for why salinomycin treatment may result in reduced transferrin uptake despite increased levels of transferrin receptor.
- provide some information about the chemistry behind the capacity of these molecules to promote iron accumulation in the lysosome. Are they iron chelators? or simply affect the function of organelles where iron is trafficked? Reply: It is now stated in the Introduction that salinomycin and ironomycin interact with iron(II) (see page 2, lines 60-66).
- comment on the pKa of salinomycin and ironomycin. Are they lysosomotropic agents? Ironomycin has a secondary amine with a pKa close to 10, and hence, might readily accumulate in the lysosome, but is difficult to infer this for salinomycin. Please comment on this. Reply: It is now mentioned in the Introduction that salinomycin and ironomycin are lysosomotropic agents (see page 2, lines 66-67). Also, we have commented on how the pKa of the ionophores affects their lysosomal accumulation (see page 2, lines 67-69).
Minor points
-at several points in the text absolute GI50 values (with standard errors) obtained under different experimental conditions (Fe replenishment, transferrin, Vit E) are compared for both compounds. A Table summarizing these results may help to compare the full set of data. In some cases, the comparisons can be made using relative comparisons (i.e. fold or % increase/decrease of compound potenty). Reply: As suggested by the reviewer, a table summarising the GI50 values has been included in the manuscript (see page 3, Table 1, and lines 105-108).
- the methodological details in the figures' captions are a bit too long and redundant with those described in the Mat&Met section. Thus, it would be convenient to shorten the captions. Reply: We have shortened the figure legends of Figures 2, 4, and 6. However, we found it difficult to shorten the figure legends of Figures 3 and 5 as they didn’t contain extensive methodological details.
- Caption Fig. 2: include "not shown" for the condition treated with only 0.9% DMSO. Reply: Rather than indicating “not shown”, we have created a figure that shows the cell swelling in the presence of 0.9% DMSO. This figure is shown in Appendix A as Figure A1 (see page 13, Figure A1, and lines 491-493; the previous Figure A1 has been changed to Figure A2). This is mentioned in the figure legend of Figure 2 (see page 3, lines 96-97).
- line 78: remove one dash from "...C20--propargylamine...." Reply: The second dash has been removed (see page 3. line 103).
- delete "in Bloodstream-Form Trypanosomes" in all subheadings, because at several points of the manuscript, it is stated that the results presented are restricted to performed on this stage and organism. Reply: As suggested by the reviewer, the subheadings have been accordingly shortened (see page 4, line 123, and page 6, line 220).
- line 287: indicate whether the T. b. brucei cell line used corresponds to a strain with capacity (pleomorphic) or not (monomorphic) to complete the full differentiation cycle of the parasite. Reply: It is now stated that the T. brucei cell line used in the study is a monomorphic strain (see page 11, line 382).
- line 325 or 163: indicate why a protease inhibitor was included in the assay (I guess to avoid transferrin degradation). Reply: It is now mentioned that the protease inhibitor was included in the assay to prevent the degradation of transferrin (see page 11, lines 420-421).
- line 365-366: indicate in % the decrease in viability excerted by 50 uM ferric iron. Reply: It is now stated in the legend of Figure A2 that 50 μM ammonium ferric citrate decreases the viability of trypanosomes by 10% (see page 13, lines 500-501).
Round 2
Reviewer 2 Report
Comments and Suggestions for Authors
In the manuscript: "Increased trypanocidal activity of the salinomycin derivative ironomycin is due to ROS production and iron uptake impairment", the authors aimed to elucidate the mechanisms behind the trypanocidal effects of salinomycin and ironomycin on bloodstream-form Trypanosoma brucei. They concentrated on the ionophoretic activity of these compounds, osmotic water absorption, their ROS production capability, and their impact on iron uptake.
In this revised manuscript, the authors not only addressed the suggestions provided but also made significant improvements in clarifying the objectives, experimental design, and obtained results. This resulted in a clearer explanation of the trypanocidal mechanisms of action of ironomycin on the parasite's bloodstream forms.
However, it is recommended that the authors:
1) In line 404-405, “After a further 48 h of incubation, the absorbance of each well at 570 nm (test wavelength) and 630 nm (reference wavelength) was read on a BioTek ELx808 microplate reader.”
The resazurin incubation conditions used (48 hours) have already been documented in a previous study by the research group, which also references an earlier manuscript (Merschjohann, K., Sporer, F., Steverding, D., & Wink, M. (2001). In vitro effect of alkaloids on bloodstream forms of Trypanosoma brucei and T. congolense. Planta Medica, 67(7), 623–627. https://doi.org/10.1055/s-2001-17351) where only 24 hours of incubation with the vital dye were employed. Typically, the ideal incubation window for resazurin is between 1-5 hours. For this reason, it is recommended that the authors verify these incubation times to ensure the technique’s optimal sensitivity. While this vital dye does not harm cells, prolonged reaction times may occasionally result in signal saturation.
2) It is recommended to include the ironomycin curve (without co-treatment) in Figure 5 to make the conclusion more evident.
3) In line 300-304. “The subcellular localization of the fluorescent spot is consistent with the position of the lysosome in bloodstream-form trypanosomes [23,24]. This result suggests that iron(II) is not transported out of the lysosome but sequestered in the organelle. A similar finding was also made for trypanosomes incubated with salinomycin, although the intensity of the fluorescent spotseems to be slightly weaker (Figure 7, SAL).”
It is recommended that the authors rephrase the sentence to soften the tone of the conclusions, as there is no quantification to support the difference in labeling intensity between the signal observed in cells treated with ironomycin and salinomycin. Additionally, including a colocalization image with a lysosome marker would have been ideal.
Beyond the suggestions, this manuscript offers a clearer insight into the trypanocidal mechanisms of action of ironomycin and salinomycin on the bloodstream form of Trypanosoma. At this point, I believe this work could make a significant contribution to both the development and the understanding of the mechanisms of action of new trypanocidal drugs for African trypanosomiasis.
Author Response
In the manuscript: "Increased trypanocidal activity of the salinomycin derivative ironomycin is due to ROS production and iron uptake impairment", the authors aimed to elucidate the mechanisms behind the trypanocidal effects of salinomycin and ironomycin on bloodstream-form Trypanosoma brucei. They concentrated on the ionophoretic activity of these compounds, osmotic water absorption, their ROS production capability, and their impact on iron uptake.
In this revised manuscript, the authors not only addressed the suggestions provided but also made significant improvements in clarifying the objectives, experimental design, and obtained results. This resulted in a clearer explanation of the trypanocidal mechanisms of action of ironomycin on the parasite's bloodstream forms. Reply: We thank the reviewer for their effort to have critically reviewed the manuscript again.
However, it is recommended that the authors:
1) In line 404-405, “After a further 48 h of incubation, the absorbance of each well at 570 nm (test wavelength) and 630 nm (reference wavelength) was read on a BioTek ELx808 microplate reader.”
The resazurin incubation conditions used (48 hours) have already been documented in a previous study by the research group, which also references an earlier manuscript (Merschjohann, K., Sporer, F., Steverding, D., & Wink, M. (2001). In vitro effect of alkaloids on bloodstream forms of Trypanosoma brucei and T. congolense. Planta Medica, 67(7), 623–627. https://doi.org/10.1055/s-2001-17351) where only 24 hours of incubation with the vital dye were employed. Reply: This reference was removed during the first revision as it does not describe the exact method used for determining the trypanocidal activity. Instead, the reference “Steverding, D., HuczyÅ„ski, A. Trypanosoma brucei: trypanocidal and cell swelling activity of lasalocid acid. Parasitol. Res. 2017, 116, 3229-3233” is cited which describes the correct method. Typically, the ideal incubation window for resazurin is between 1-5 hours. Reply: The short incubation time for resazurin can be used when measuring fluorescence. However, we measured absorbance which requires a much longer incubation time so that enough blue/violet resazurin can be converted into pink resorufin. The fact that we measured absorbance was clearly stated in the method section and is indicated in the figure y-axis titles. For this reason, it is recommended that the authors verify these incubation times to ensure the technique’s optimal sensitivity. While this vital dye does not harm cells, prolonged reaction times may occasionally result in signal saturation. Reply: The resazurin assay as described in the manuscript (24h pre-incubation before adding resazurin followed by another 48 h incubation before measuring the absorbance) is an established method that I have used since 2009 for determining the trypanocidal activity of compounds in several studies that have been published in prestigious journals (e.g. Eur. J. Med. Chem., Molecules, J. Nat. Med., and Dalton Trans.).
2) It is recommended to include the ironomycin curve (without co-treatment) in Figure 5 to make the conclusion more evident. Reply: As suggested by the reviewer, we have included the ironomycin curve without co-treatment in Figure 5 (see page 7, Fig. 5 and line 263).
3) In line 300-304. “The subcellular localization of the fluorescent spot is consistent with the position of the lysosome in bloodstream-form trypanosomes [23,24]. This result suggests that iron(II) is not transported out of the lysosome but sequestered in the organelle. A similar finding was also made for trypanosomes incubated with salinomycin, although the intensity of the fluorescent spot seems to be slightly weaker (Figure 7, SAL).” It is recommended that the authors rephrase the sentence to soften the tone of the conclusions, as there is no quantification to support the difference in labeling intensity between the signal observed in cells treated with ironomycin and salinomycin. Reply: As suggested by the reviewer we have softened the statement regarding the intensity of the signal observed in cells treated with salinomycin (see pages 8-9, lines 304-308). Additionally, including a colocalization image with a lysosome marker would have been ideal. Reply: There is no need to perform colocalization staining with a lysosome marker since the lysosome of bloodstream forms of T. brucei is a single vacuole situated just posterior to the centrally located nucleus. The position of the fluorescent spot is identical to that of the lysosome. The position of the fluorescent spot has been more precisely described (see page 8, line 299).
Beyond the suggestions, this manuscript offers a clearer insight into the trypanocidal mechanisms of action of ironomycin and salinomycin on the bloodstream form of Trypanosoma. At this point, I believe this work could make a significant contribution to both the development and the understanding of the mechanisms of action of new trypanocidal drugs for African trypanosomiasis. Reply: We hope the reviewer is satisfied with the additional revision of the manuscript.
Reviewer 3 Report
Comments and Suggestions for Authors
The authors addressed very satisfactorily all the concerns raised in my revision, and congratulate tham for the work done.
Author Response
The authors addressed very satisfactorily all the concerns raised in my revision, and congratulate them for the work done. Reply: We thank the reviewer for their support of our manuscript.